# On a Variational and Convex Model of the Blake–Zisserman Type for Segmentation of Low-Contrast and Piecewise Smooth Images

**DOI:** 10.3390/jimaging7110228

**Published:** 2021-10-28

**Authors:** Liam Burrows, Anis Theljani, Ke Chen

**Affiliations:** Liverpool Centre of Mathematics for Healthcare and Centre for Mathematical Imaging Techniques, Department of Mathematical Sciences, University of Liverpool, Liverpool L69 7ZL, UK; l.r.burrows@liv.ac.uk (L.B.); theljani@liverpool.ac.uk (A.T.)

**Keywords:** image segmentation, Mumford–Shah, Blake–Zisserman, game theory

## Abstract

This paper proposes a new variational model for segmentation of low-contrast and piecewise smooth images. The model is motivated by the two-stage image segmentation work of Cai–Chan–Zeng (2013) for the Mumford–Shah model. To deal with low-contrast images more effectively, especially in treating higher-order discontinuities, we follow the idea of the Blake–Zisserman model instead of the Mumford–Shah. Two practical ideas are introduced here: first, a convex relaxation idea is used to derive an implementable formulation, and second, a game reformulation is proposed to reduce the strong dependence of coupling parameters. The proposed model is then analysed for existence and further solved by an ADMM solver. Numerical experiments can show that the new model outperforms the current state-of-the-art models for some challenging and low-contrast images.

## 1. Introduction

Image segmentation is a widely studied yet still challenging subject, especially for new and emerging imaging modalities where Mumford–Shah and extremely strong noise may be present. Of course, extremely simple images with clear contrast, without noise and without blur may be segmented by the simple methods, such as thresholding the image intensity values.

Real-life images inevitably have noise and low contrast which poses a challenge for the simple algorithms. Variational segmentation models generally provide more robust solutions for complex images and can usually be categorised loosely into two categories: edge-based or region-based models. Well-known edge-based methods include Kass et al. [1] and Caselles et al. [2]. Region-based models are generally referred to the pioneering work of Mumford–Shah (MS) [3], with some simplified variants such as Chan–Vese [4,5] that are most widely used.

In the last few years, when mentioning segmentation of challenging images, we would automatically recommend machine-learning-based approaches such as the UNet [6] and Resnet [7]. However, such works are data-dependent, and often, networks are tailored to a specific task. Firstly, they require training data which may not be available (or reliably available) at all. Secondly, we cannot yet conduct automatic transfer learning from a subject area to another to overcome the lack of sufficient training data, e.g., aircraft identification network cannot be adapted to identification of livers in medical imaging. A reliable way of overcoming the lack of sufficient training data is by weakly or semi-supervised learning which uses a small set of training data (in a supervised way) and a larger set of data without annotations (in an unsupervised way) [8,9]. Here, ‘unsupervised’ means that a suitable segmentation model is required; developing such a model is our aim.

This paper addresses the fundamental problem of how to segment low-contrast images where image features of interest have piecewise smooth intensities. In fact, the difficulties of the two problems, namely low-contrast and piecewise smooth features, are well-known challenges. Low contrast implies that edge information by way of image gradients alone is not sufficient enough to detect weak jumps. Moreover, many well-known models such as [4] or its variants assume an input image has approximately piecewise constant intensities; piecewise smooth features imply these models cannot segment such features (or a feature would be split into sub-regions (or multiple phases) according to the intensity distribution, which means that the segmentation is already incorrect). Many approximation models based on the MS [3] can deal with segmentation of piecewise smooth features but not necessarily images displaying low contrast.

Therefore, this paper considers the Blake–Zisserman model [10] which can improve on the MS model [3]. The model [10] cannot be implemented directly and exactly, just as with the MS [3], which was never solved directly.

The rest of the paper is organised as follows. Section 2 briefly reviews related segmentation models. Section 3 introduces our new model and a game theory reformulation to facilitate subsequent solutions. Proof of the solution existence of the game formulation is given. Section 4 presents our numerical algorithm for the game formulation, and Section 5 shows numerical experiments. Brief conclusions are drawn in Section 6.

## 2. Related Works

The above-mentioned Mumford–Shah model [3] minimises the following:(1)$$\begin{array}{c} F_{MS}\left(g,\Gamma \right)=\frac{\lambda }{2}\int_{\Omega }{\left(f-g\right)}^{2}dx+\int_{\Omega \backslash \Gamma }{\left|\nabla g\right|}^{2}dx+\nu H^{1}\left(\Gamma \right), \end{array}$$
given the input (possibly noisy) image $f:\Omega \to R^{2}$, where, most importantly, the segmentation is defined by the unknown boundary Γ, $g:\Omega \to R^{2}$ is a piecewise smooth approximation of *f*, and $H^{1}\left(\Gamma \right)$ denotes the Hausdorff measure (i.e., length of the boundary). In the literature, there are many follow-up works of this paper, proposed to make revised models implementable numerically. Successful results have been obtained. See [11,12,13], among others.

However, for images that have weak edges possibly buried in noise and blur, the Mumford–Shah type models may fail to capture the ‘discontinuities of second kind’ or gradient discontinuity, which may be called the staircasing effect for gradients. The Blake–Zisserman (BZ) type model [10], though less well-known and published earlier than [3], can be very useful for a class of challenging images where MS is less effective; e.g., see [14,15]. The functional of a BZ model takes the form
(2)$$\begin{array}{c} F_{BZ}\left(g,\Gamma ,\Gamma_{\nabla }\right)=\frac{\lambda }{2}\int_{\Omega }{\left(f-g\right)}^{2}dx+\int_{\Omega \backslash (\Gamma ⋃\Gamma_{\nabla })}{\left|\nabla^{2}g\right|}^{2}dx+\nu_{1}H^{1}\left(\Gamma \right)++\nu_{2}H^{1}\left(\Gamma_{\nabla }\backslash \Gamma \right), \end{array}$$
where g,∇g∈BV(Ω). Here, $\Gamma_{\nabla }$ is the discontinuity of ∇g. As with the original formulation (Equation 1), the BZ model (Equation 2) is theoretical, not in a readily solvable form. This paper will propose an approximate and solvable model.

Our work is motivated by Cai–Chan–Zeng [12], who derived a solvable and convex model for (Equation 1). We now review this model briefly. As a first step of reformulation of (Equation 1), Cai–Chan–Zeng [12] rewrites (Equation 1) in an equivalent form
(3)$$\begin{array}{cc} \;\;E(\Sigma ,g_{1},g_{2})= & \frac{\lambda }{2}\int_{\Omega \backslash \Sigma }{\left(f-g_{2}\right)}^{2}dx+\frac{\mu }{2}\int_{\Omega \backslash \Sigma }{\left|\nabla g_{2}\right|}^{2}dx\;+\\ \; & \frac{\lambda }{2}\int_{\Sigma \backslash \Gamma }{\left(f-g_{1}\right)}^{2}dx+\frac{\mu }{2}\int_{\Sigma \backslash \Gamma }{\left|\nabla g_{1}\right|}^{2}dx+H^{1}\left(\Gamma \right), \end{array}$$
where Γ is assumed to be a Jordan curve as the boundary ∂Σ for the closed domain $\Sigma =\bar{Inside(\Gamma )}$. Hence, $g_{1},g_{2}$ are defined in the inside and outside of Γ, respectively. Of course, both $g_{1},g_{2}$ can be smoothly extended to the entire domain Ω. A key observation in [12], motivated by [5], is that the term $H^{1}\left(\Gamma \right)$, which is the length of Γ, may be approximated by $\int_{\Omega }\left|\nabla g_{1}\right|dx$. Then, viewing the smooth functions $g_{1},g_{2}$ as a single function, the model by [12] is the following:(4)$$\begin{array}{c} \min_{g}\frac{\lambda }{2}\int_{\Omega }{\left(f-g\right)}^{2}dx+\frac{\mu }{2}\int_{\Omega }{\left|\nabla g\right|}^{2}dx+\int_{\Omega }\left|\nabla g\right|dx. \end{array}$$

We now propose a solvable model based on the Blake–Zisserman model (Equation 2). Assume the given image is *f*, and our approximation is $g\in W^{1,2}\left(\Omega \right)$, with $\nabla g\in {\left(W^{1,2}\left(\Omega \right)\right)}^{2}$.

Motivated by the work of [12], we shall respectively approximate the key quantities $H^{1}\left(\Gamma \right),H^{1}\left(\Gamma_{\nabla }\backslash \Gamma \right)$ by $\int_{\Omega }|\nabla g|dx,\int_{\Omega }\left|\nabla \left(\nabla g\right)\right|dx$. Therefore, our initial minimisation model takes the form
(5)$$\begin{array}{cc} E_{BZ}\left(g\right)= & \frac{\lambda }{2}\int_{\Omega }{\left(g-f\right)}^{2}dx+\frac{\nu_{1}}{2}\int_{\Omega }|\nabla^{2}{g|}^{2}dx+\frac{\nu_{2}}{2}\int_{\Omega }{\left|\nabla g\right|}^{2}dx\\ \; & \;+\int_{\Omega }|\nabla^{2}g|dx+\int_{\Omega }\left|\nabla g\right|dx. \end{array}$$

While (Equation 5) is well-defined in terms of solvability, to facilitate the choice of coupling parameters, we now consider a game formulation. A game formulation encourages independent players to complete with each. Here, each player is a sub-problem in an optimisation formulation; see [16]. Here, independence means that parameters of sub-problems do not have to rely on each other.

## 3. The New Model and Reformulation as a Nash Game

In this work, we are interested in a particular case of a two-player game formulation. Instead of optimising the single energy (Equation 5), we consider a game reformulation, where two individuals, or ‘players’, are involved. The first player is the variable *g*, and the second one will be introduced by using the idea of operator splitting [17] to reduce the high-order derivatives in (Equation 5) as first-order terms and to simplify subsequent solution. The solution to this game is the Nash equilibrium, whose existence must be established. For important techniques and results in game theory and its connections to partial differential equations (PDEs) for other problems, the reader is directed to [18,19,20,21].

More precisely, let G be an approximation for vector ∇g. Then, we propose our new model, approximating (Equation 5), as
(6)$$\begin{array}{cc} \min_{g}E_{1}\left(g,G\right) & =\left\{\frac{\lambda_{1}}{2}||f-{g||}_{2}^{2}+\frac{\mu_{1}}{2}{\left||\nabla g|\right|}_{2}^{2}+{\left||\nabla g|\right|}_{1}+\frac{\xi_{1}}{2}||G-{\nabla g||}_{2}^{2}\right\}, \end{array}$$
(7)$$\begin{array}{cc} \min_{G}E_{2}\left(G,g\right) & =\left\{\frac{\lambda_{2}}{2}||\nabla f-{G||}_{2}^{2}+\frac{\mu_{2}}{2}{\left||\nabla G|\right|}_{2}^{2}+{\left||\nabla G|\right|}_{1}+\frac{\xi_{2}}{2}||G-{\nabla g||}_{2}^{2}\right\}, \end{array}$$
where $g\in W^{1,2}\left(\Omega \right)$ and $G\in {\left(W^{1,2}\left(\Omega \right)\right)}^{2}$.

**Definition** **1.**
*A pair $\left(g^{*},G^{*}\right)$ in the space $W=W^{1,2}\left(\Omega \right)\times {\left(W^{1,2}\left(\Omega \right)\right)}^{2}$ is called a Nash equilibrium for the game involving the two energies $E_{1}\left(\cdot \right)$ and $E_{2}\left(\cdot \right)$, defined on W, if *

$$\left\{\begin{array}{cc} & E_{1}\left(g^{*},G^{*}\right)\leq E_{1}\left(g,G^{*}\right),\;\forall \;g\in W^{1,2}\left(\Omega \right),\\ & E_{2}\left(g^{*},G^{*}\right)\leq E_{2}\left(g^{*},G\right),\;\forall \;G\in {\left(W^{1,2}\left(\Omega \right)\right)}^{2}. \end{array}\right.$$



One could consider only the single energy $E_{1}+E_{2}$ to be optimised; however, for the theoretical analysis, the ellipticity of the sum energy is not guaranteed because of the coupling term between *g* and G. Hence, the existence of minimisers is not straightforward. However, we emphasise that in the game formulation, the energies $E_{1}\left(\cdot ,G\right)$ and $E_{2}\left(g,\cdot \right)$ are partially elliptic, i.e., with respect to the variables *g* and G, respectively. This is a very important property which eases the proof of the existence of Nash equilibrium.

**Proposition** **1.**
*There exists a unique Nash equilibrium $\left(g^{*},G^{*}\right)\in W^{1,2}\left(\Omega \right)\times {\left(W^{1,2}\left(\Omega \right)\right)}^{2}$ for the two-player game involving the costs functional $E_{1}\left(\cdot ,\cdot \right)$ and $E_{2}\left(\cdot ,\cdot \right)$ in (Equation 6) and (7).*


**Proof** **of** **Proposition** **1.**Since
$E_{1}\left(\cdot ,G\right)$ is partially strict convex, partially elliptic and weakly lower semi-continuous with respect to variable *g*,$E_{2}\left(g,\cdot \right)$ is partially strict convex, partially elliptic and weakly lower semi-continuous with respect to variable G,
the proof is a straightforward and direct application of the the Nash theorem [22]. □

## 4. Numerical Algorithms and Implementation

In this section, we detail the numerical algorithm to solve our game model and show how we utilise the outputs to obtain a segmentation result.

### 4.1. Stage One: Solution of the Main Model Using ADMM

The discretised version of our two-player game model (Equation 6) and (7) is given as follows:$$\begin{array}{cc} \; & \min_{g}\left\{\frac{\lambda_{1}}{2}||f-{g||}_{2}^{2}+\frac{\mu_{1}}{2}{\left||\nabla g|\right|}_{2}^{2}+{\left||\nabla g|\right|}_{1}+\frac{\xi_{1}}{2}||G-{\nabla g||}_{2}^{2}\right\},\\ \; & \min_{G}\left\{\frac{\lambda_{2}}{2}||\nabla f-{G||}_{2}^{2}+\frac{\mu_{2}}{2}{\left||\nabla G|\right|}_{2}^{2}+{\left||\nabla G|\right|}_{1}+\frac{\xi_{2}}{2}||G-{\nabla g||}_{2}^{2}\right\}, \end{array}$$
where ${\left||\nabla g|\right|}_{1}=\sum_{i\in \Omega }\sqrt{{\left(\nabla_{x}g\right)}_{i}^{2}+{\left(\nabla_{y}g\right)}_{i}^{2}}$ and ${\left||\nabla G|\right|}_{1}=\sum_{i\in \Omega }\left|{\left(\nabla_{x}G_{x}\right)}_{i}+{\left(\nabla_{y}G_{y}\right)}_{i}\right|$. The gradient operator $\nabla =(\nabla_{x},\nabla_{y})$ is discretised using backwards differences with zero Neumann boundary conditions.

We aim to solve the coupled problem using the split-Bregman variant of the alternating direction method of multipliers (ADMM) [23], which is commonly used for problems containing $L_{1}$ regularisation. In order to do this, we introduce a new variable into each sub-problem:$$\begin{array}{cc} \; & \min_{g,v}\left\{\frac{\lambda_{1}}{2}||f-{g||}_{2}^{2}+\frac{\mu_{1}}{2}{\left||\nabla g|\right|}_{2}^{2}+{\left||v|\right|}_{1}+\frac{\xi_{1}}{2}||G-{\nabla g||}_{2}^{2}\right\},\;\;\mathrm{such}\;\mathrm{that}\;v=\nabla g,\\ \; & \min_{G,w}\left\{\frac{\lambda_{2}}{2}||\nabla f-{G||}_{2}^{2}+\frac{\mu_{2}}{2}{\left||\nabla G|\right|}_{2}^{2}+{\left||w|\right|}_{1}+\frac{\xi_{2}}{2}||G-{\nabla g||}_{2}^{2}\right\},\;\;\mathrm{such}\;\mathrm{that}\;w=\nabla G. \end{array}$$

Applying split-Bregman to enforce the constraints gives us:$$\begin{array}{cc} \; & \min_{g,v}\left\{\frac{\lambda_{1}}{2}||f-{g||}_{2}^{2}+\frac{\mu_{1}}{2}{\left||\nabla g|\right|}_{2}^{2}+{\left||v|\right|}_{1}+\frac{\xi_{1}}{2}||G-{\nabla g||}_{2}^{2}+\frac{\rho_{1}}{2}||v-\nabla g-b_{1}{\left|\right|}_{2}^{2}\right\},\\ \; & \min_{G,w}\left\{\frac{\lambda_{2}}{2}||\nabla f-{G||}_{2}^{2}+\frac{\mu_{2}}{2}{\left||\nabla G|\right|}_{2}^{2}+{\left||w|\right|}_{1}+\frac{\xi_{2}}{2}||G-{\nabla g||}_{2}^{2}+\frac{\rho_{2}}{2}||w-\nabla G-b_{2}{\left|\right|}_{2}^{2}\right\}. \end{array}$$

We detail briefly how to solve each of the sub-problems:

***g* sub-problem:** We aim to solve the minimisation problem for fixed $v^{\left(k\right)},G^{\left(k\right)},b_{1}^{\left(k\right)}$:$$g^{\left(k+1\right)}=\arg\;\min_{g}\left\{\frac{\lambda_{1}}{2}||f-{g||}_{2}^{2}+\frac{\mu_{1}}{2}{\left||\nabla g|\right|}_{2}^{2}+\frac{\xi_{1}}{2}\left|\right|G^{\left(k\right)}-{\nabla g||}_{2}^{2}+\frac{\rho_{1}}{2}\left|\right|{\hat{v}}^{\left(k\right)}-\nabla g{\left|\right|}_{2}^{2}\right\},$$
where ${\hat{v}}^{\left(k\right)}=v^{\left(k\right)}-b_{1}^{\left(k\right)}$, which amounts to solving the following equation for *g*:$$\left(\lambda_{1}+\left(\mu_{1}+\xi_{1}+\rho_{1}\right)\nabla^{2}\right)g=\lambda_{1}f+\xi_{1}\nabla^{T}G^{\left(k\right)}+\rho_{1}\nabla^{T}\left(v^{\left(k\right)}-b_{1}^{\left(k\right)}\right).$$

This can be solved by using discrete Fourier transforms F:(8)$$g^{\left(k+1\right)}=F^{-1}\left(\frac{F(L)}{M}\right)$$
where
$$L=\lambda_{1}f+\xi_{1}\nabla^{T}G^{\left(k\right)}+\rho_{1}\nabla^{T}\left(v^{\left(k\right)}-b_{1}^{\left(k\right)}\right),\;\;M=\lambda_{1}+\left(\mu_{1}+\xi_{1}+\rho_{1}\right)F\left(\nabla^{2}\right).$$

**v sub-problem:** We aim to solve this minimisation problem for fixed $g^{\left(k+1\right)},b_{1}^{\left(k\right)}$:$$v^{\left(k+1\right)}=\arg\min_{v}\left\{{\left||v|\right|}_{1}+\frac{\rho_{1}}{2}||v-\nabla g^{\left(k+1\right)}-b_{1}^{\left(k\right)}{\left|\right|}_{2}^{2}\right\},$$
which is solved analytically by a generalised shrinkage formula:(9)$$v_{x}^{\left(k+1\right)}=\max\left(s^{\left(k\right)}-\frac{1}{\rho_{1}},0\right)\frac{s_{x}^{\left(k\right)}}{s^{\left(k\right)}},\;v_{y}^{\left(k+1\right)}=\max\left(s^{\left(k\right)}-\frac{1}{\rho_{1}},0\right)\frac{s_{y}^{\left(k\right)}}{s^{\left(k\right)}},$$
where $s_{1}^{\left(k\right)}=\nabla_{x}g^{\left(k+1\right)}+{\left(b_{1}^{\left(k\right)}\right)}_{x}$, $s_{2}^{\left(k\right)}=\nabla_{y}g^{\left(k+1\right)}+{\left(b_{1}^{\left(k\right)}\right)}_{y}$ and



$s^{\left(k\right)}=\sqrt{{\left(s_{1}^{\left(k\right)}\right)}^{2}+{\left(s_{2}^{\left(k\right)}\right)}^{2}}.$



The associated Bregman update is:(10)$${\left(b_{1}^{\left(k+1\right)}\right)}_{x}={\left(b_{1}^{\left(k\right)}\right)}_{x}+\nabla_{x}g^{\left(k\right)}-v_{x}^{\left(k+1\right)},\;{\left(b_{1}^{\left(k+1\right)}\right)}_{y}={\left(b_{1}^{\left(k\right)}\right)}_{y}+\nabla_{y}g^{\left(k\right)}-v_{y}^{\left(k+1\right)}.$$

**G sub-problem:** We aim to solve the minimisation problem for fixed $g^{\left(k+1\right)},w^{\left(k\right)},b_{2}^{\left(k\right)}$:$$G^{\left(k+1\right)}=\arg\min_{G}\{\frac{\lambda_{2}}{2}||\nabla f-{G||}_{2}^{2}+\frac{\mu_{2}}{2}{\left||\nabla G|\right|}_{2}^{2}+$$
$$\;\frac{\xi_{2}}{2}||G-\nabla g^{\left(k+1\right)}{\left|\right|}_{2}^{2}+\frac{\rho_{2}}{2}\left|\right|w^{\left(k\right)}-\nabla G-b_{2}^{\left(k\right)}{\left|\right|}_{2}^{2}\},$$
whose solution is defined by the following:$$\left(\lambda_{2}+\xi_{2}+\left(\mu_{2}+\rho_{2}\right)\nabla^{2}\right)G=\lambda_{2}\nabla f+\xi_{2}\nabla g^{\left(k+1\right)}+\rho_{2}\nabla^{T}\left(w^{\left(k\right)}-b_{2}^{\left(k\right)}\right).$$
To find the solution G, we apply discrete Fourier transforms F:(11)$$G_{x}^{\left(k+1\right)}=F\left(\frac{F(N_{1})}{P}\right),\;G_{y}^{\left(k+1\right)}=F\left(\frac{F(N_{2})}{P}\right),$$
where $P=\lambda_{2}+\xi_{2}+\left(\mu_{2}+\rho_{2}\right)F\left(\nabla^{2}\right)$, $N_{1}=\nabla_{x}f+\xi_{2}\nabla_{x}g^{\left(k+1\right)}+\rho_{2}\nabla_{x}^{T}\left(w^{\left(k\right)}-b_{2}^{\left(k\right)}\right)$, and $N_{2}=\nabla_{y}f+\xi_{2}\nabla_{y}g^{\left(k+1\right)}+\rho_{2}\nabla_{y}^{T}\left(w^{\left(k\right)}-b_{2}^{\left(k\right)}\right)$.

***w* sub-problem:** We aim to solve the minimisation problem for fixed $G^{\left(k+1\right)},b_{2}^{\left(k\right)}:$$$w^{\left(k+1\right)}=\arg\min_{w}\left\{{\left||w|\right|}_{1}+\frac{\rho_{2}}{2}||w-\nabla G^{\left(k+1\right)}-b_{2}^{\left(k\right)}{\left|\right|}_{2}^{2}\right\},$$
which, similar to (Equation 9), is solved by using a shrinkage formula:(12)$$w^{\left(k+1\right)}=\max\left(|r^{\left(k\right)}|-\frac{1}{\rho_{2}},0\right)\frac{r^{\left(k\right)}}{|r^{\left(k\right)}|},$$
where $r^{\left(k\right)}=\nabla G^{\left(k+1\right)}+b_{2}^{\left(k\right)}$.

### 4.2. Stage Two: Segmentation of *f* by Thresholding *g*

In order to acquire a segmentation result for *f*, we take the minimiser *g* from stage one and threshold it according to some suitably defined threshold parameter(s). As in [12], the advantage of this method is that changing the threshold value(s) does not require the re-computation of the optimisation done in stage one.

There are two strategies that can be employed to define the threshold(s). The first is to use the k-means algorithm, which is an automatic method that partitions a given input into *K* clusters, for K≥2. The second is to define the threshold value(s) manually, which generally provides better results. As the threshold values are applied after optimisation, a wide range of values can easily be tried and the best selected. In our experiments, we use manual threshold values for two-phase segmentation, whereas for multiphase segmentation with multiple threshold values, we use k-means to simplify the process.

## 5. Numerical Results

In this section, we display some examples of the performance of our model and compare it with a number of models, namely:CRCV: Convex relaxed Chan–Vese model [5];CCZ: The two-stage convex variant of the Mumford–Shah model by Cai et al. [12] given in (Equation 4);CNC: The convex non-convex segmentation by Chan et al. [24];T-ROF: The T-ROF model by Cai et al. [25];
and also a deep learning model.

We first show some visual comparisons, where noise is added to the original image, and then later do a quantitative analysis on a dataset. Note that all the models above (and ours) except for the CRCV model is capable of multiphase segmentation, whereas the CRCV model (in the Chan–Vese framework) is only capable of two-phase segmentation. For this reason, in the experiments, we only include the CRCV model in two-phase examples.

### 5.1. Qualitative Results

In Figure 1, we show an image from an ultrasound. We add additive Gaussian noise with mean 0 and standard deviation 10. We display the outputs of all the competing models, the segmentation result overlaid on the original image, and for all but the CRCV show the segmentation result after thresholding (as the segmentation result after thresholding is the binary output shown first). We see that the segmentation result from our model is better at segmenting the object in the image, noticing that our segmentation effectively segments the “tail” part at the top of the object, whereas the CCZ model fails to segment it well. The CRCV and CNC models segment the tail but fail to remove the noise. We note that the T-ROF model is the best competing model but does not quite segment all the tail.

Similarly, in Figure 2, we show another two-phase segmentation example, where we have the clean image but add Gaussian noise with mean 0 and standard deviation 25. It is clear that none of the competing models are as good as ours. Our result manages to preserve more detail in general, notably at the strand at the top, and the curved structure at the bottom of the image, without being susceptible to the noise.

In Figure 3, Figure 4 and Figure 5, we show some examples of multiphase segmentation on MRI images of the brain. In all cases, we add Gaussian noise with mean 0 and standard deviation 17 and run the noisy image as input to both for all models but the CRCV model (as this is a two-phase model only). The output is then given as input to the k-means algorithm with K=4. We show the clustering output in the final column of the relevant figures. We see that the segmentation result of our model is better at finding some of the finer edges; for example, the white matter segmentation from our model is in general more detailed than the segmentation from the competing ones.

### 5.2. Quantitative Analysis

To assess our method quantitatively, we run our model on 20 images in the Digital Retinal Images for Vessel Extraction (DRIVE) dataset (https://drive.grand-challenge.org/ accessed 25 October 2021). We use the manual segmentation image as the clean image and add additive Gaussian noise with mean 0 and standard deviation 100 to use as the input image, as shown in Figure 6, Figure 7, Figure 8 and Figure 9a,b respectively. We display the output of the competing models and our model here as well as a deep learning model (abbreviated as DL). We trained a U-Net [6] network on 15 of the images (and used the other five as validation set), where the noisy image served as input and we trained with binary cross-entropy loss function to match with the clean image. The results are good; however, we lack a large dataset to provide the impressive result that deep learning approaches usually provide.

Figure 6, Figure 7, Figure 8 and Figure 9 are four examples on the given dataset; however, we run on the 20 images available to provide some quantitative analysis. We use the *DICE* coefficient and the *JACCARD* similarity coefficient as quantitative measures to evaluate the performance of segmentation results. Given a binary segmentation result Σ from a model and ground truth segmentation GT, the *DICE* coefficient is given as:$$DICE\left(\Sigma ,GT\right)=\frac{2|\Sigma \cap GT|}{\left|\Sigma |+|GT\right|}.$$

The *JACCARD* similarity coefficient is given as:$$JACCARD\left(\Sigma ,GT\right)=\frac{\left|\Sigma \cup GT\right|}{\left|\Sigma |+|GT|-|\Sigma \cap GT\right|}.$$

In Table 1, we show the mean and standard deviation values of the *DICE* and *JACCARD* scores on the dataset. We see clearly that our model is more effective than the Cai model from these results. We note that the numerical values provided for the DL method are run on all 20 images in the dataset; however, the DL was trained on 15 of these images. This is somewhat of an unfair comparison; however, we see that the numerical values for our approach are still larger than the values for the DL approach despite this. Figure 10 shows the boxplots of quantitative results on the data, for further visualisation.

## 6. Conclusions

In this paper, we have developed a convex relaxed game formulation of the less well-known Blake–Zisserman model in order to segment images with low contrast and strong noise. The advantages of the game formulation are that the existence of Nash equilibrium can be proved and there is less dependence on parameters for each sub-problem, i.e., parameters of each sub-problem do not rely on each other, and so can be tuned appropriately and separately. The game model was implemented using a fast split-Bregman algorithm, and numerical experiments show improvements in segmentation results over competing models, especially over the well-known Mumford–Shah type methods for low-contrast images.

## Figures and Tables

**Figure 1 jimaging-07-00228-f001:**
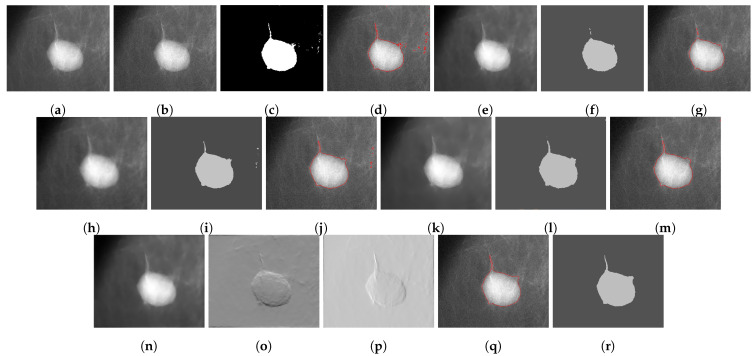
Results from an ultrasound image: (**a**) Clean image. (**b**) Noisy image used as input to the models. (**c**) Output of the CRCV model. (**d**) CRCV contour. (**e**) Output of CCZ. (**f**) CCZ after thresholding. (**g**) CCZ contour. (**h**) Output of CNC. (**i**) CNC after thresholding. (**j**) CNC contour. (**k**) Output of T-ROF. (**l**) T-ROF after thresholding. (**m**) T-ROF contour. (**n**) Output *g* of our model. (**o**) Output $G_{x}$ of our model. (**p**) Output $G_{y}$ of our model. (**q**) Ours after thresholding. (**r**) Our contour.

**Figure 2 jimaging-07-00228-f002:**
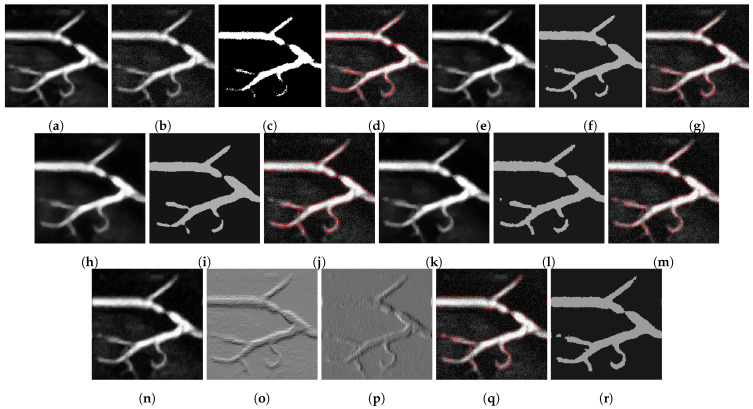
Results from a blood vessel image: (**a**) Clean image. (**b**) Noisy image used as input to the models. (**c**) Output of the CRCV model. (**d**) CRCV contour. (**e**) Output of CCZ. (**f**) CCZ after thresholding. (**g**) CCZ contour. (**h**) Output of CNC. (**i**) CNC after thresholding. (**j**) CNC contour. (**k**) Output of T-ROF. (**l**) T-ROF after thresholding. (**m**) T-ROF contour. (**n**) Output *g* of our model. (**o**) Output $G_{x}$ of our model. (**p**) Output $G_{y}$ of our model. (**q**) Ours after thresholding. (**r**) Our contour.

**Figure 3 jimaging-07-00228-f003:**
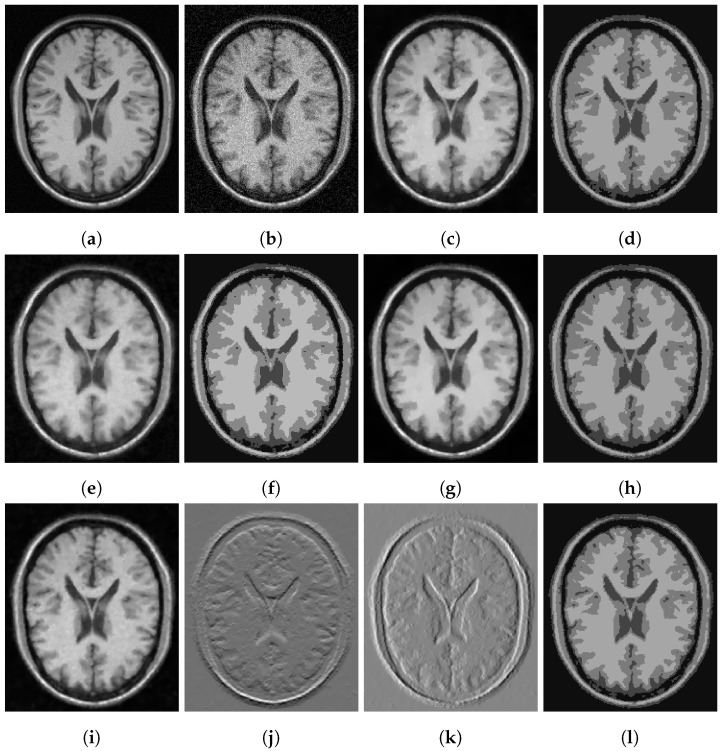
MRI segmentation: (**a**) Clean image. (**b**) Noisy image used as input to the models. (**c**) Output of CCZ. (**d**) CCZ after thresholding. (**e**) Output of CNC. (**f**) CNC after thresholding. (**g**) Output of T-ROF. (**h**) T-ROF after thresholding. (**i**) Output *g* of our model. (**j**) Output $G_{x}$ of our model. (**k**) Output $G_{y}$ of our model. (**l**) Ours after thresholding.

**Figure 4 jimaging-07-00228-f004:**
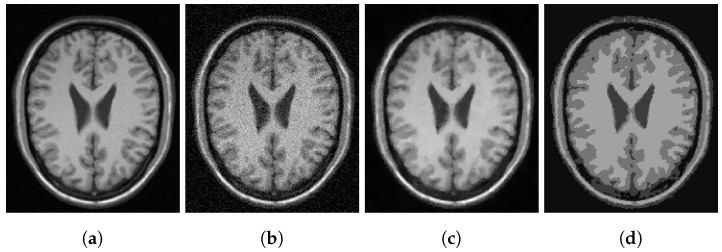
MRI segmentation: (**a**) Clean image. (**b**) Noisy image used as input to the models. (**c**) Output of CCZ. (**d**) CCZ after thresholding. (**e**) Output of CNC. (**f**) CNC after thresholding. (**g**) Output of T-ROF. (**h**) T-ROF after thresholding. (**i**) Output *g* of our model. (**j**) Output Gx of our model. (**k**) Output Gy of our model. (**l**) Ours after thresholding.

**Figure 5 jimaging-07-00228-f005:**
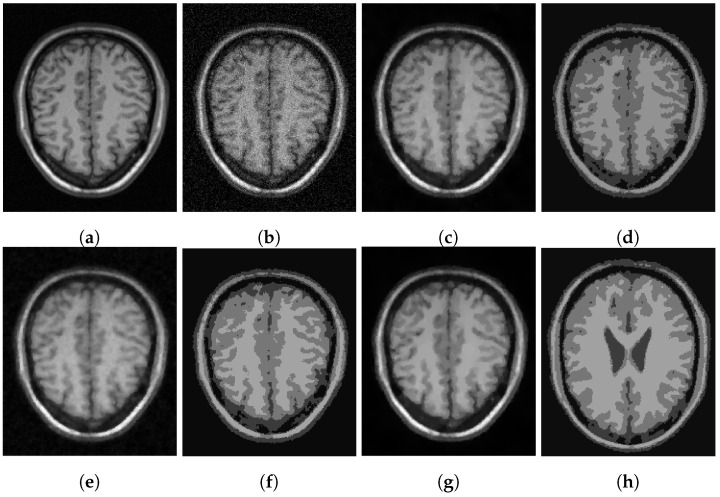
MRI segmentation: (**a**) Clean image. (**b**) Noisy image used as input to the models. (**c**) Output of CCZ. (**d**) CCZ after thresholding. (**e**) Output of CNC. (**f**) CNC after thresholding. (**g**) Output of T-ROF. (**h**) T-ROF after thresholding. (**i**) Output *g* of our model. (**j**) Output Gx of our model. (**k**) Output Gy of our model. (**l**) Ours after thresholding.

**Figure 6 jimaging-07-00228-f006:**
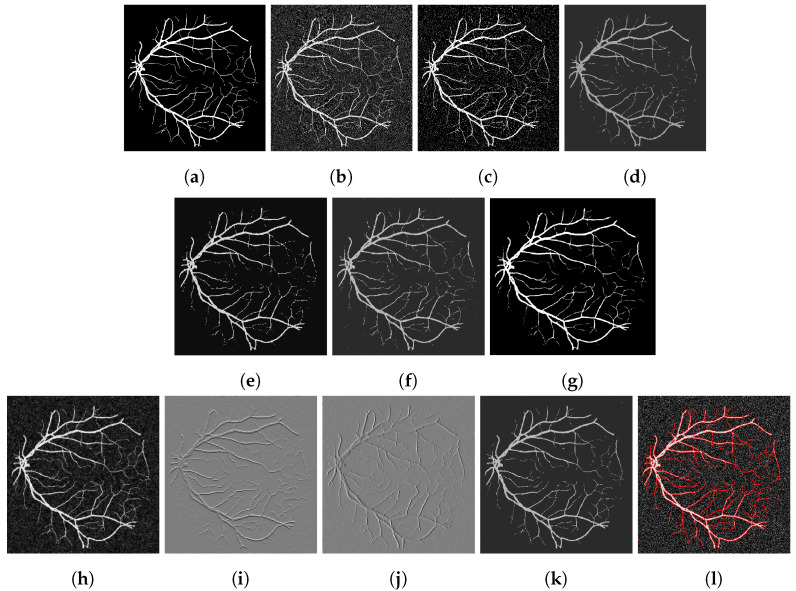
(**a**) Clean image. (**b**) Noisy image used as input to the models. (**c**) Output of the CRCV model. (**d**) CCZ after thresholding. (**e**) CNC after thresholding. (**f**) T-ROF after thresholding. (**g**) DL Output. (**h**) Output *g* of our model. (**i**) Output $G_{x}$ of our model. (**j**) Output $G_{y}$ of our model. (**k**) Ours after thresholding. (**l**) Our contour.

**Figure 7 jimaging-07-00228-f007:**
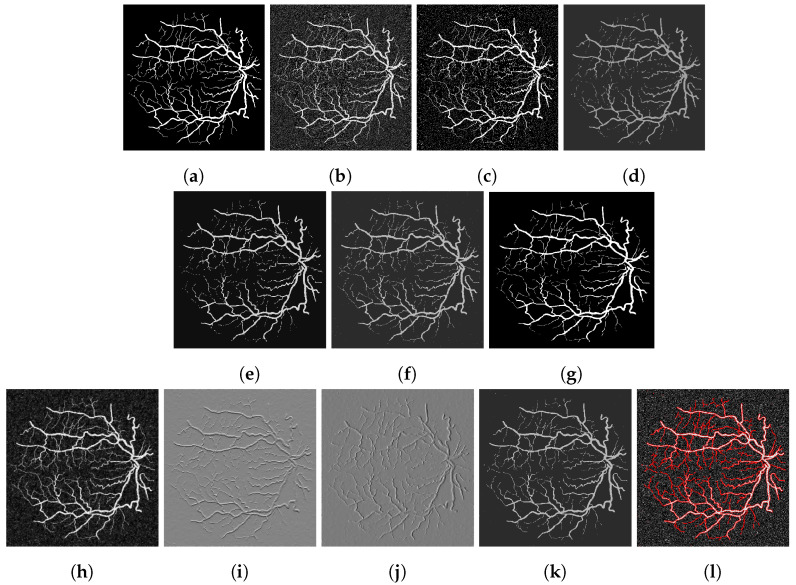
(**a**) Clean image. (**b**) Noisy image used as input to the models. (**c**) Output of the CRCV model. (**d**) CCZ after thresholding. (**e**) CNC after thresholding. (**f**) T-ROF after thresholding. (**g**) DL Output. (**h**) Output *g* of our model. (**i**) Output $G_{x}$ of our model. (**j**) Output $G_{y}$ of our model. (**k**) Ours after thresholding. (**l**) Our contour.

**Figure 8 jimaging-07-00228-f008:**
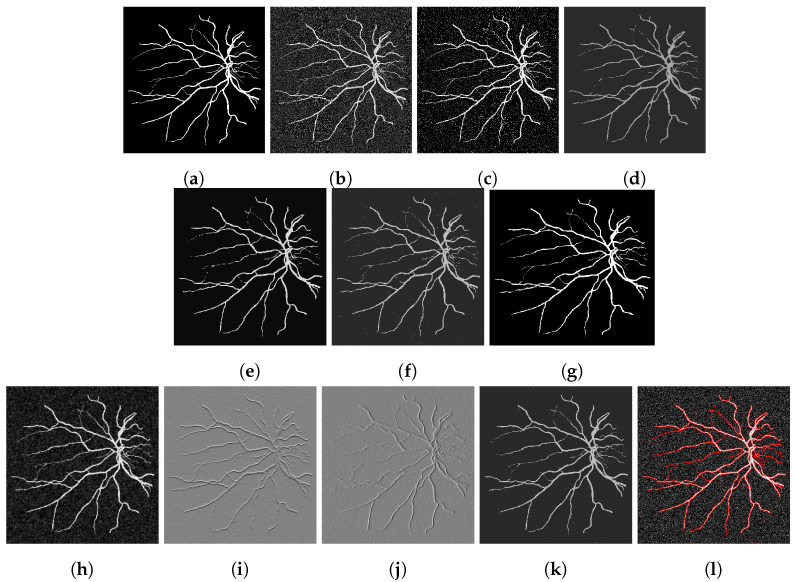
(**a**) Clean image. (**b**) Noisy image used as input to the models. (**c**) Output of the CRCV model. (**d**) CCZ after thresholding. (**e**) CNC after thresholding. (**f**) T-ROF after thresholding. (**g**) DL Output. (**h**) Output *g* of our model. (**i**) Output $G_{x}$ of our model. (**j**) Output $G_{y}$ of our model. (**k**) Ours after thresholding. (**l**) Our contour.

**Figure 9 jimaging-07-00228-f009:**
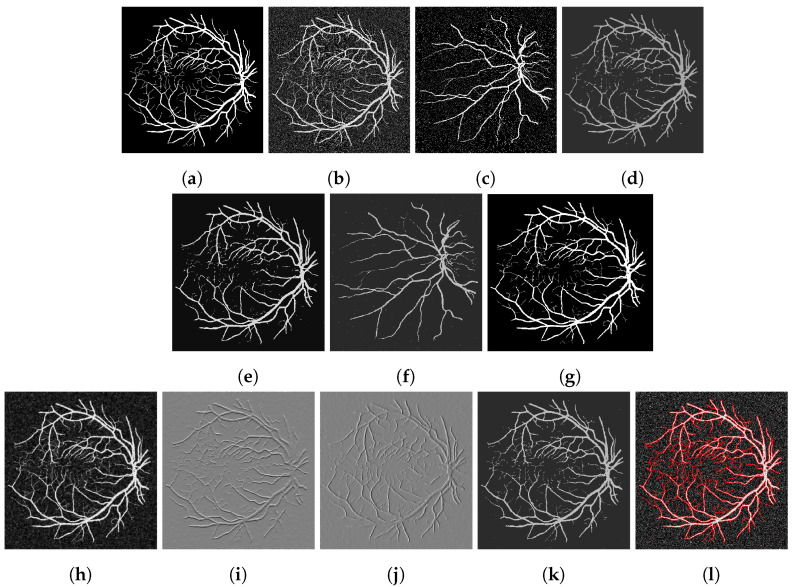
(**a**) Clean image. (**b**) Noisy image used as input to the models. (**c**) Output of the CRCV model. (**d**) CCZ after thresholding. (**e**) CNC after thresholding. (**f**) T-ROF after thresholding. (**g**) DL Output. (**h**) Output *g* of our model. (**i**) Output $G_{x}$ of our model. (**j**) Output $G_{y}$ of our model. (**k**) Ours after thresholding. (**l**) Our contour.

**Figure 10 jimaging-07-00228-f010:**
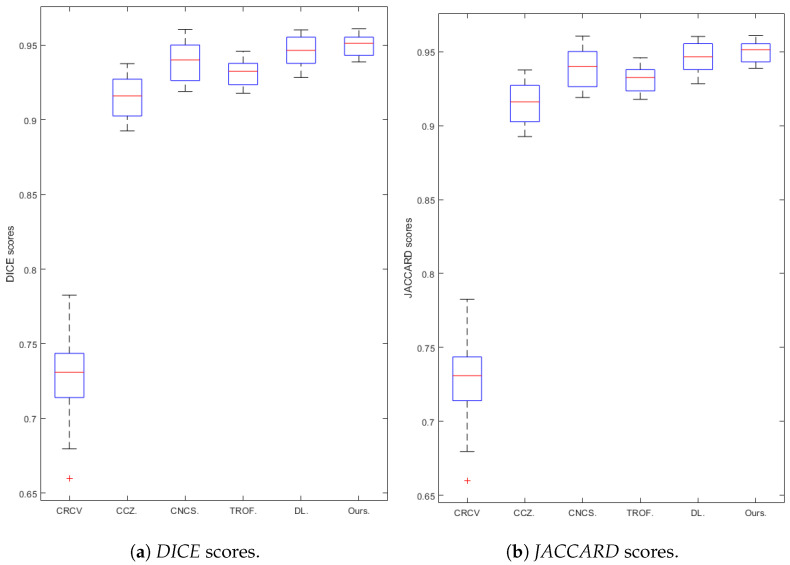
Comparison of six methods (CRCV, CCZ, CNCS, TROF, DL, Ours): box plots of the quantitative results for *DICE* (**a**) and *JACCARD* (**b**) scores.

**Table 1 jimaging-07-00228-t001:** Quantitative results from images in the DRIVE dataset. Here, we show the two methods evaluated on 20 images and display the mean and standard deviations of both the *DICE* coefficient and *JACCARD* score. Note that the DL method was trained on 15 of these 20 images.

	*DICE*		*JACCARD*	
	μ	σ	μ	σ
CRCV	0.727	0.0291	0.573	0.0358
CCZ	0.914	0.0139	0.843	0.0236
CNC	0.939	0.0131	0.884	0.0233
T-ROF	0.932	0.0083	0.872	0.0145
DL	0.946	0.0091	0.898	0.0163
Ours	0.950	0.0073	0.905	0.0133

## Data Availability

The Digital Retinal Images for Vessel Extraction (DRIVE) dataset can be found at https://drive.grand-challenge.org/. Data accessed: 25 October 2021.

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
