# Peer review of "On a Variational and Convex Model of the Blake–Zisserman Type for Segmentation of Low-Contrast and Piecewise Smooth Images"

_2313-433X, 2021, doi:10.3390/jimaging7110228_

Round 1

Reviewer 1 Report

The proposed work looks interesting which involves the game theory. However, the paper has several issues.

Firstly, some mathematical formulas are incorrect or out of the paper width. In (2), $\Gamma_\Delta$ is not defined or explained. In (5), the third and fourth terms are almost the same except the front coefficient. It is unclear in which sense (5) is well defined. In Definition 1, the first inequality seems problematic. In Sec. 4.1, when applying ADMM to solve the (g,v)-/(G,w)-subproblems, why does the variable v/w only appear in one of terms in the objective function? 

Is Stage 2 necessary and how to choose the thresholds in practice? The proposed method intends to improve Cai's work with limited theoretical novelty. Numerical comparisons should also include some other state-of-the-art segmentation methods.

Lastly, substantial English revisions should be made to improve the paper quality. The Cai's model was cited in many places and could be termed with an abbreviation. 

Reviewer 2 Report

This paper provides a convex relaxed game formulation of the Blake-Zisserman model in order to segment images with low contrast and strong noise. The advantages of the game formulation is the existence of Nash equilbrium can be proved and there is less dependence on parameters for each sub-problem. Numerical experiments show that the new model outperforms the current state of art models for some challenging and low contrast images.

The results are interesting and relevant for the researchers on this subject. However, there are some minor problems that need improvement. For that, please see the attached file.

Author Response

We have made the changes highlighted by reviewer 2. We would like to thank them for pointing out various typographical errors. Some key errors highlighted which we have fixed is the text of proposition one, and as suggested we have implemented abbreviations for the competing models. 

Round 2

Reviewer 1 Report

The authors have made satisfactory revisions. I do not have any further comments.